# The cervicovaginal microbiome associates with spatially restricted host transcriptional signatures throughout the human ectocervical epithelium and submucosa

Vilde Kaldhusdal[1,2*], Mathias Franzén Boger[1,2], Adam D. Burgener[1,2,3,4], Julie Lajoie[5,6], Kenneth Omollo[5,7], Joshua Kimani[5,7], Annelie Tjernlund[1,2], Keith R. Fowke[5,6,7,8], Douglas S. Kwon[9], Gabriella Edfeldt[1,2], Kristina Broliden[1,2]

1 Department of Medicine Solna, Division of Infectious Diseases, Karolinska Institutet, Stockholm, Sweden, 2 Department of Infectious Diseases, Karolinska University Hospital, Center for Molecular Medicine, Stockholm, Sweden, 3 Center for Global Health and Diseases, Department of Pathology, Case Western Reserve University School of Medicine, Cleveland, Ohio, United States of America, 4 Department of Obstetrics, Gynecology and Reproductive Sciences, Max Rady College of Medicine, University of Manitoba, Winnipeg, Canada, 5 Department of Medical Microbiology and Infectious Diseases, University of Manitoba, Winnipeg, Canada, 6 Department of Medical Microbiology and Immunology, University of Nairobi, Nairobi, Kenya, 7 Partners for Health and Development in Africa, Nairobi, Kenya, 8 Department of Community Health Sciences, University of Manitoba, Winnipeg, Canada, 9 Ragon Institute of MGH, MIT and Harvard, Cambridge, Massachusetts, United States of America

* vilde.kaldhusdal@ki.se

## Abstract

The cervicovaginal microbiome is a key biological determinant of human immunodeficiency virus (HIV) susceptibility, but its underlying impact on the ectocervical transcriptional landscape is unclear. Ectocervical tissue samples from Kenyan female sex workers were categorized into pre-defined cervicovaginal microbiome groups based on dominant compositions: *Lactobacillus crispatus/acidophilus*, *Lactobacillus iners*, *Gardnerella*, and 'highly diverse'. The tissue samples (n=21) were assessed using spatial transcriptomics, revealing three epithelial, one mixed border, and nine submucosal gene clusters. Differential gene expression analysis across the microbiome groups and gene clusters identified 3,771 unique genes. The highly diverse microbiome group associated with the largest differences, mostly located near the epithelial basal membrane, encompassing genes involved in epithelial maintenance, submucosal extracellular matrix structures, and immune function. The *L. crispatus/acidophilus*-dominated group was identified by genes involved in active immune engagement, supporting mucosal barrier integrity. Weighted gene co-expression analysis confirmed tissue-wide altered gene expression associated with all microbiome groups and with individual bacterial taxa. Despite the assumption that microbiome colonization is restricted to the luminal surface, the transcriptional landscape was affected throughout the mucosa, with the most pronounced effect near both

which permits unrestricted use, distribution, and reproduction in any medium, provided the original author and source are credited.

**Data availability statement:** 16S rRNA sequencing data used to define the study groups from paired cervicovaginal lavage samples were previously deposited in the European Nucleotide Archive (accession number PRJEB50325). The clinical characteristics of the study participants and the spatial transcriptomics data are available from the GEO public repository (accession ID GSE217237). The raw sequencing data from the study participants cannot be held in a public repository due to the sensitive nature of such personal data. Requests for data access can be made to the Karolinska Institutet Research Data Office (contact: rdo@ki.se), and access will be granted if the request meets the requirements of the data policy. All code related to this paper are available at https://github.com/vildeka/Spatial_Microbiota.

**Funding:** Funding was provided by the Swedish Research Council (https://www.vr.se), (VR-MH 2022-01001, K.B.), the Swedish Physicians Against AIDS Research Foundation (http://www.aidsfond.se), (FOb2023-0014, VK), and the Canadian Institutes of Health Research (https://cihr-irsc.gc.ca), (CIHR, MOP #86721, K.R.F.). The project also received funding from the European Union's Horizon 2020 Research and Innovation programme (https://research-and-innovation.ec.europa.eu/index_en) under grant agreement no. 847943 (MISTRAL; K.B., A.D.B.). The funders had no role in study design, data collection and analysis, decision to publish, or preparation of the manuscript.

**Competing interests:** The authors have declared that no competing interests exist.

sides of the basal membrane. This broad association with the mucosal barrier integrity could affect susceptibility to HIV acquisition.

## Author summary

The microorganisms that live in and on our bodies—collectively known as the microbiome—can influence our health in many ways, including how our immune system behaves and how susceptible we may be to infections. In this study, we focused on the human cervix, a key part of the reproductive tract, and investigated how different types of vaginal bacteria might influence the activity of genes in this tissue. By combining microbiome profiling with a powerful technique called spatial transcriptomics, we were able to see how gene activity varies across different regions of the cervix and how it associated with the surrounding microbes. We found that certain bacterial communities are linked to changes in gene expression deep within the tissue, not just on the surface. These findings help us understand how microbial communities might contribute to health risks like infection or inflammation and may one day inform strategies to prevent conditions such as HIV or adverse pregnancy outcomes. Our work shows the importance of looking beyond surface-level changes and considering how microbes influence whole tissues in complex ways.

## Introduction

Adolescent girls and young women account for a significant proportion of new human immunodeficiency virus (HIV) infections in sub-Saharan Africa. The 15–29-year age group is 2.4 times more likely to contract HIV than their male counterparts. Several factors contribute to this vulnerability, including structural and biological factors and early sexual activity [1–3]. Among biological factors, the cervicovaginal microbiome composition significantly impacts a woman's susceptibility to HIV and other sexually transmitted infections [4–6]. An optimal cervicovaginal microbiome is typically dominated by *Lactobacillus* species, which help maintain an acidic environment that protects against genital infections, such as HIV [4]. However, an anaerobic, highly diverse microbiome dominated by non-*Lactobacillus* bacteria is more commonly observed in women in sub-Saharan Africa [7]. This microbial profile was associated with a 4-fold increase in HIV transmission risk in a cohort of young South African women [4]. A highly diverse cervicovaginal microbiome is associated with local inflammation, which compromises the local mucosal barrier and increases the number of activated mucosal CD4$^+$ T cells [8]. A non-*Lactobacillus*-dominated microbiome can also affect the efficacy of HIV prevention methods, such as vaginal microbicides and oral pre-exposure prophylaxis [9,10].

Spatial transcriptomics offers several advantages for the study of microbiome-associated mucosal alterations in tissue. Unlike traditional transcriptomic assessment

of bulk tissue samples, spatial transcriptomics maintains the spatial arrangement of cells within tissues [11,12]. Spatial transcriptomics further provides a topographical map of gene expression patterns across tissue sections. Such spatial resolution is important for understanding molecular processes in the ectocervix, which consists of morphologically and functionally distinct compartments. The squamous epithelium provides a barrier against mechanical stress and microbial invasion. It consists of defined cell layers of differentiating keratinocytes. The submucosa is an underlying loose connective tissue rich in fibroblasts, collagen, and elastin fibers, along with blood vessels, lymphatics, and immune cells.

Spatial transcriptomics has been applied to human cervical tissue samples in the context of human papillomavirus-related cancers, hormonal contraceptive use, and HIV infection [13–18]. In two cross-sectional clinical studies, we performed transcriptional analysis of cytobrush-derived endocervical cells from South African young women, and bulk transcriptomics analysis of ectocervical tissue samples from Kenyan female sex workers, respectively. We revealed that different cervicovaginal microbiome communities were correlated with distinct host gene expression signatures [19,20]. These findings are expanded here by applying spatial transcriptomics on a subset of the Kenyan ectocervical tissue samples. This technique allows a significantly higher spatial resolution and reveals novel functional gene expression pathways associated with the microbiome, as well as a broad distribution of the gene signatures throughout the mucosa.

## Results

### Sociodemographic, clinical, and reposited experimental data of study participants

This study includes spatial transcriptomics analyses of ectocervical tissue samples that previously underwent bulk transcriptomic analysis [19]. The samples were collected as part of our broader longitudinal investigation within the Sex Worker Outreach Program in Pumwani, Nairobi, Kenya [21,22].

The previously published 16S rRNA V4 gene sequencing data of corresponding cervicovaginal lavage samples [19] were used to select samples (Fig 1A), aiming to include at least four samples per pre-defined cervicovaginal microbiome composition-based study group [19]. High RNA integrity (RNA integrity number ≥7) and a morphology suitable for data visualization were further used as sample selection criteria, resulting in 21 tissue samples. Samples with more than 80% relative abundance of *L. crispatus*/*acidophilus* and/or *L. jensenii* were defined as L1 ('*L. crispatus*', n = 4). Samples with >80% *Lactobacillus* spp., mainly *L. iners*, were assigned to L2 ('*L. Iners*', n = 6); Samples with >10% *Gardnerella* and <5% *Prevotella* were assigned to L3 ('*Gardnerella*', n = 6); Samples with >5% *Prevotella* were assigned to L4 ('highly diverse', n = 5) (S1 Fig). In our previous study, the natural prevalence of the L-groups was as follows: L1: 9%; L2: 28%, L3: 19%, L4: 36% [19].

Participants using the hormonal contraceptive compound depot medroxyprogesterone acetate or were HIV seropositive were excluded. The sample selection aimed to represent the follicular phase of the menstrual cycle, although a few participants did not meet this criterion. These and other clinical and sociodemographic characteristics are presented at the group level (Table 1).

### Gene expression clustering reveals epithelial stratification and submucosal heterogeneity

We first characterized host molecular changes associated with different microbiome profiles using spatial transcriptomics (S2 Fig). Unsupervised clustering was applied to group spots according to gene expression [23]. The clustering resulted in three epithelial (Clusters 5, 6, and 7), one mixed border (Cluster 8), and nine submucosal (Clusters 0–4, 9–12) clusters (Figs 1B and S3). The three epithelial clusters were labeled as *Superficial*, *Upper Intermediate*, and *Lower Intermediate* layers, corresponding to clusters 5, 6, and 7, respectively. For simplicity, the mixed-border cluster (Cluster 8), which contained both epithelial and submucosal spots, is referred to as the *Basal* layer.

The top marker genes for each cluster, identified through one-vs-rest differential expression analysis, largely aligned with their spatial location within the tissue architecture (Fig 1C). For instance, the epithelial clusters had genes

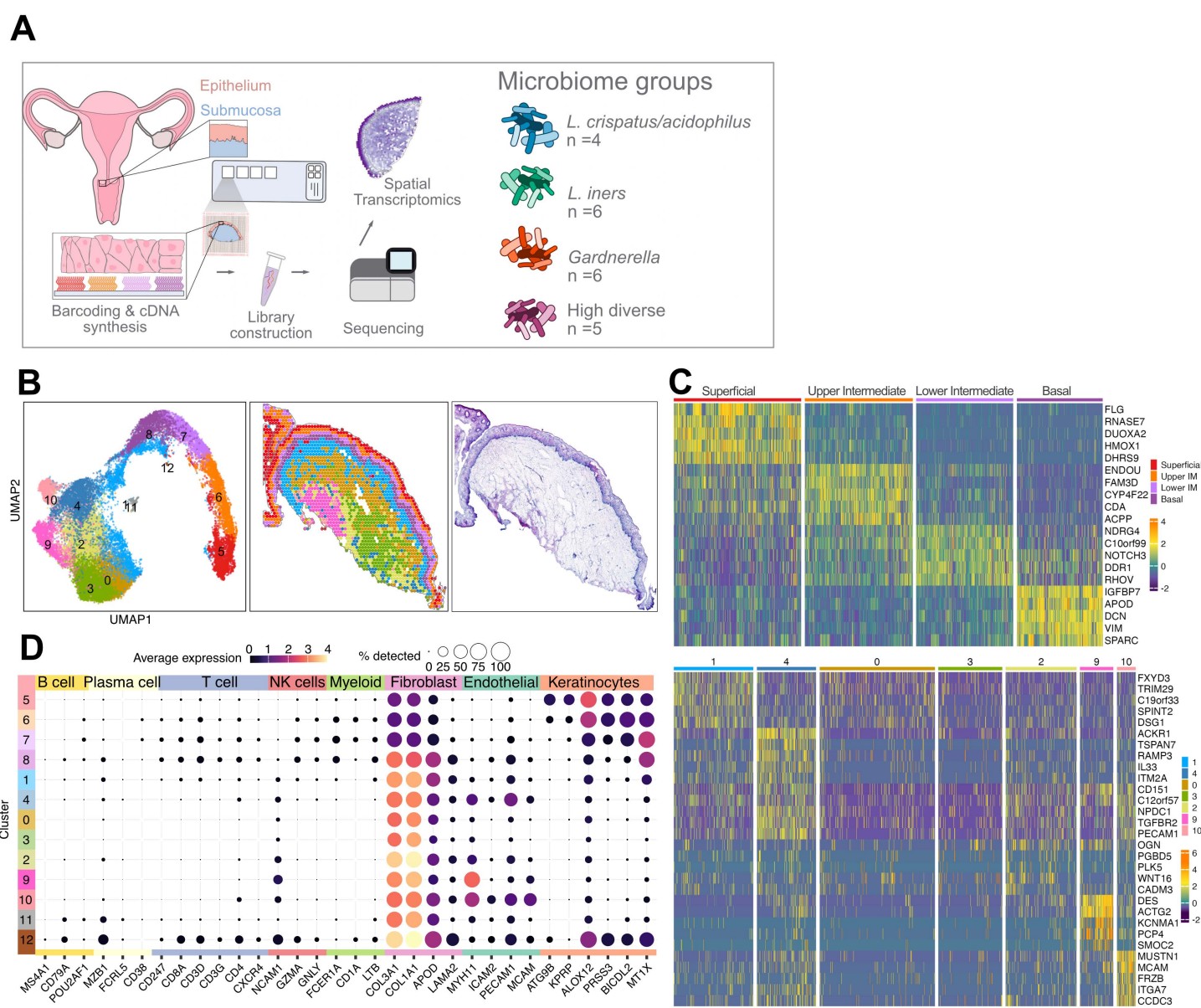

**Fig 1. Overview of study design and spatial gene expression linked to ectocervical structures and cell-type markers. A** Ectocervical biopsies (n = 21) were selected for spatial transcriptomics based on previously published 16S rRNA V4 gene sequencing data on paired cervicovaginal lavage samples. The samples collectively represent four distinct microbiome profiles, which were compared for differences in host-gene expression. Differentially expressed genes were further examined by weighted co-expression analysis. **B** The mRNA expression was mapped within individual "spots" evenly distributed across the tissue. Each spot measures 55 µm in diameter, estimated to represent 1–10 cells. The resulting dataset comprising tissue samples from 21 study participants included 30,427 spots, with a median of 2,319 total transcripts and 1,265 genes per spot. Spots from all samples were classified by unsupervised clustering visualized by Uniform Manifold Approximation and Projection (UMAP) (left) and on tissue (sample P014) (middle), HE image (right). 13 clusters (3 epithelial, 1 mixed and 9 submucosal) are depicted. **C** Top 5 marker genes per cluster, across the two morphological structures (epithelium and submucosa). **D** Average expression reflects the mean log-normalized expression across cells of selected cell marker genes, visualized for each of the 13 clusters (cluster 0-12).

PLOS Pathogens

**Table 1. Sociodemographic data and clinical characteristics of study participants.**

| | L1 n=(4) | L2 n=(6) | L3 n=(6) | L4 n=(5) | na |
|---|---|---|---|---|---|
| | **Median (range or %)** | | | | |
| *Sociodemographic parameters* | | | | | |
| Age (years) | 38 (30-47) | 35 (25–38) | 42 (20-48) | 36 (25-43) | 0 |
| Time in sex work (months) | 36 (24-120) | 60 (24-120) | 114 (2-372) | 24 (18-144) | 0 |
| Number of weekly clients | 8 (4–18) | 5 (3-50) | 4 (0-5) | 4 (3-50) | 1 |
| Children (number) | 3 (2–4) | 2 (1–4) | 2 (1–4) | 2 (0-4) | 2 |
| Educational level (years in school) | 9 (8–12) | 11 (8–14) | 8 (7–10) | 11 (7–21) | 0 |
| Marital status (married) | 2 (50%) | 0 | 2 (33%) | 2 (40%) | 0 |
| *Sex hormone status* | | | | | |
| Time since onset of menses (days) | 29 (9-44) | 7 (5–14) | 8 (5–19) | 6 (3–12) | 3 |
| Estradiol (pg/mL) | 221 (86-296) | 77 (22-242) | 107 (22-405) | 45 (27-290) | 0 |
| Below LLD (22 pg/ml) | 0 | 1 (17%) | 1 (17%) | 0 | 0 |
| DMPA use | 0 | 0 | 0 | 0 | 0 |
| *STIs and vaginal health* | | | | | |
| HIV pos | 0 | 0 | 0 | 0 | 0 |
| Presence of NG | 0 | 0 | 0 | 0 | 0 |
| Presence of CT | 0 | 0 | 0 | 0 | 0 |
| Nugents' score (bacterial vaginosis) | | | | | |
| Normal (0–3) | 3 (75%) | 6 (100%) | 4 (67%) | 0 | 0 |
| Intermediate (4–6) | 1 (25%) | 0 | 2 (34%) | 2 (40%) | 0 |
| BV (7–10) | 0 | 0 | 0 | 3 (60%) | 0 |

Data n/a: Data not available for number of samples.

Number of weekly clients: Data from the questionnaire two weeks prior to sample collection: "How many clients did you have the past week?"

Time since onset of last menses (days).

Having an ongoing STI at time of enrolment was an exclusion criterium for participating in the study. None of the study participants were further diagnosed with *C.trachomatis* or *N. gonorroheae* at time of sample collection.

representing terminal keratinocyte differentiation and retinoid metabolism in the Superficial layer, cell differentiation in the Upper Intermediate layer, immune response and fatty acid barrier establishment in the Lower Intermediate layer, and cell division and the extracellular matrix (ECM) in the Basal layer (Cluster 8: e.g., *SPARC, DCN*) (Fig 1C, top panel and S1 Table). Together, these genes represent the processes of keratinocyte differentiation and cornification, which define the important barrier function of the multi-layered squamous epithelium.

The gene expression profiles of the submucosal clusters were less discernable and displayed a less structured spatial distribution compared to the epithelial layers (Fig 1C, lower panel). Cluster 1 was consistently located adjacent to the Basal layer and was primarily characterized by genes related to inflammation. Cluster 4 showed elevated expression of immunological and endothelial genes, while Clusters 0 and 3 exhibited reduced expression of several genes, including *C12orf57,* which is ubiquitously expressed in human tissues. This suggests that these clusters represent areas with low cell density. In contrast, Cluster 2 expressed a high *C12orf57* level. Cluster 9 was typically located deep within the submucosa and was characterized by the expression of genes commonly associated with smooth muscle cells. Finally, Cluster 10 expressed endothelial marker genes (Fig 1D and S1 Table). Clusters 11 and 12 were excluded from the analysis as they were present in only two and one sample, respectively.

Together, these findings show that the ectocervical mucosa contains distinct functional compartments as defined by unique gene expression patterns. While the epithelial gene clusters followed a clear and structured organization aligned with known tissue layers, the submucosal clusters showed more varied and less predictable patterns, suggesting a greater diversity of cell types and functions. These results are consistent with our previous spatial transcriptomics studys of the ectocervix [13,14], but by including additional samples, the present work adds further weight and resolution to these observations. In doing so, it extends beyond prior transcriptomic efforts, which have largely been limited to HPV-associated lesions or cancer [15–18], by providing a more comprehensive map of healthy ectocervical tissue architecture.

## Differential gene expression across the microbiome groups reveals the highly diverse group as the most distinct

The highly diverse microbiome group (L4) was associated with the largest number of transcriptional differences compared to the other three groups, suggesting it as the most distinct in terms of gene expression. Three ways of applying differential gene expression analysis (across, pairwise and pseudo-bulk) were performed to ensure a comprehensive analysis while minimizing biases (Fig 2A and S2–S7 Tables and S1 Text). Wilcox pairwise comparison revealed that the highly diverse (L4) group exhibited the most substantial alterations in gene expression relative to the other three microbiome groups (S4A and S4B Fig), highlighting its distinct biological signature. Based on this observation, we focused our attention on pairwise comparisons with L4 (L4 versus L1, L2, and L3). For simplicity, we only analyzed nuclear-encoded genes with known or putative functional relevance, although several ribosomal and mitochondrial genes were differentially expressed among the study groups.

## Epithelial gene expression in the highly diverse group is substantially altered in the Lower Intermediate and Basal layers

Comparison of gene expression within the epithelial layers revealed 1,403, 1,078, and 1,041 unique DEGs in the pairwise comparisons between L4 and the other groups (L1, L2, and L3). The DEGs were most pronounced in the Lower Intermediate and Basal layers (Fig 2B and S2–S4 Tables).

The L4 group exhibited significant upregulation of genes related to epithelial barrier maintenance, tissue repair, and immune responses, including *SPRR, KRT,* and *KLK* genes, across the epithelial layers compared with those in L1 and L2, with the highest log fold change seen in the Superficial layer. All significant *KLK* genes were upregulated, except for *KLK13*. Many *KRT* genes were upregulated, while others were downregulated. Notably, *KRT13* was uniquely upregulated superficially but downregulated in the Basal layer. Comparisons with L3 showed broad upregulation of keratins (*KRT14/16/17/6A/6B/6C*) and protease inhibitors across the epithelium. Additional antimicrobial and immune-related genes (*SLPI, CSTA, IGHG1/4*) were also upregulated across multiple layers and groups. The upregulation of antibody genes was most pronounced in the L2 comparison, with the Upper Intermediate layer displaying the broadest set of antibody-related genes (*IGKC, IGLC1, IGHG1/2/4, IGHA1*). The Lower Intermediate and Basal layers also displayed a broad range of innate immune-related genes, including *MIF* and *FABP5,* which are linked to macrophage responses [24,25] and retinoic acid metabolism [26]. While some layers exhibited distinct gene expression profiles, there was broad overlap in genes representing epithelial maintenance and immune function across all comparisons (Fig 2C and S2–S4 Tables).

L4-associated downregulated genes also reflected changes in epithelial barrier structure, immune regulation, and metabolic processes. Notably, the antimicrobial *OLFM4* gene stood out as downregulated across all L4 comparisons and for all epithelial layers.

Other DEGs had a more localized position, including Integrins, which were downregulated across all group comparisons in the Basal layer. Similarly, transmembrane genes and desmosomal genes (*DSG1/2, DSP, PKP4*) were downregulated in the Basal and Lower Intermediate layers. The Basal layer exhibited downregulation of many collagen-related genes, predominantly in the L4 versus L3 comparison. Various immune-related genes were also downregulated,

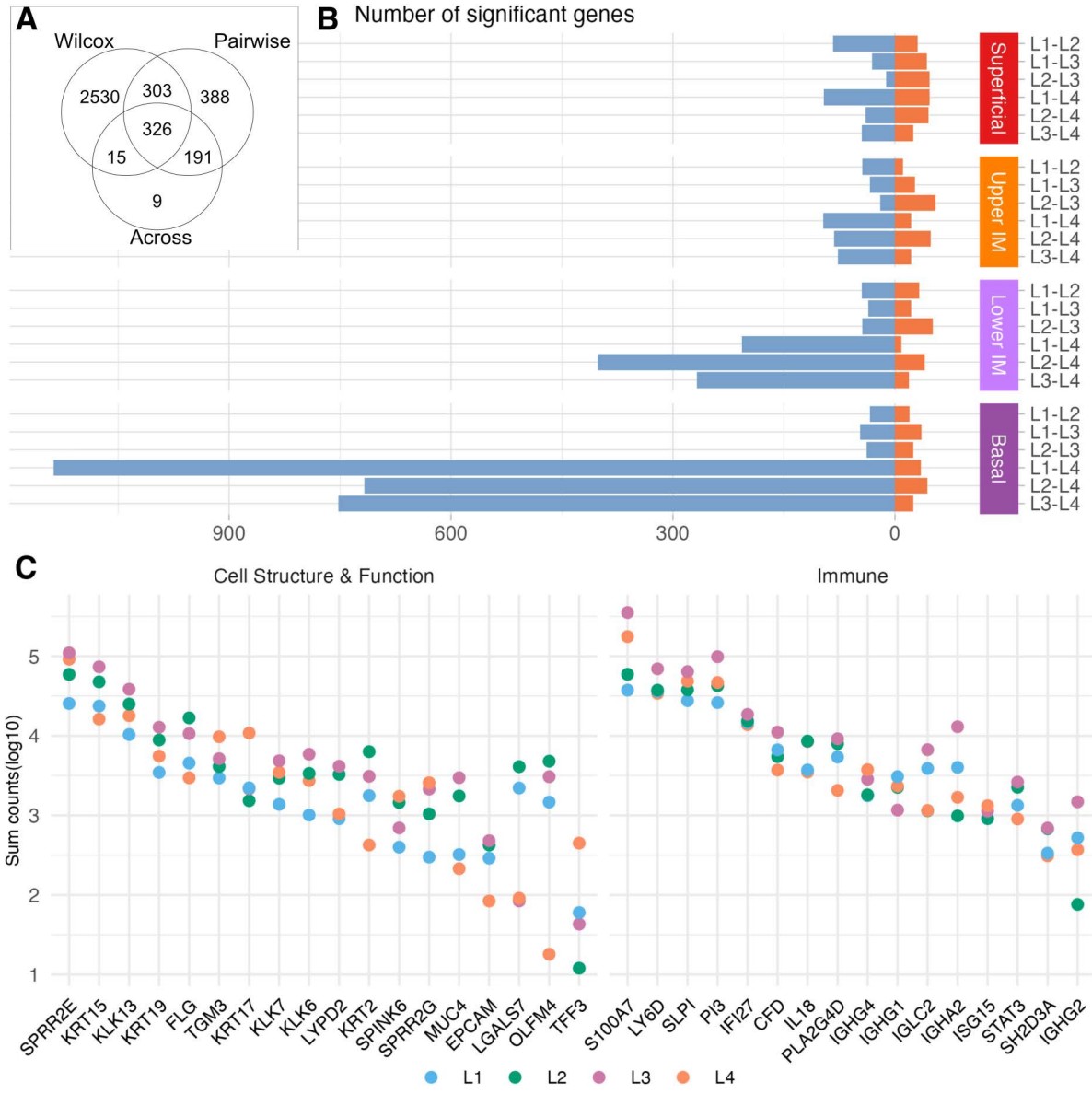

**Fig 2. Differential expression analysis overview and DEGs across epithelial layers. A** Venn diagram of the significant DEGs for the three models; pseudo DEG across, pairwise and Wilcox pairwise. **B** Stacked bar plot of all epithelial DEGs for the Wilcox pairwise model. The x-axis show number of significantly DEGs, with orange indicating upregulated and blue downregulated genes. The y-axis indicates the group comparison and the cluster. **C** Dot plot showing the $\log_{10}$ counts of selected genes summarized by microbiome groups (L1-L4) on the y-axis across the epithelial clusters (Superficial, Upper Intermediate, Lower Intermediate, and Basal).

predominantly in the Basal layer but also in the Lower Intermediate layers, including *STAT3*, which induces T-cell expansion and cytokine production, and *IL18,* which is a *STAT3* activator and is involved in tissue repair [27,28] (Fig 2B and 2C and S2–S4 Tables). Others were *HLA-C, ITGB1,* and *CD47.*

In summary, these data demonstrated both distinctly localized, as well as more widespread DEGs across the epithelium. Together, they represent the disrupted mucosal barrier integrity associated with a highly diverse microbiome. Changing perspective, the data support a protective and regulated immune environment for the *L. crispatus* group.

## Submucosal gene expression in the highly diverse group associates with alterations in immune and structural functions

Similar to the epithelium results, the L4 group displayed most DEGs within the submucosa in pairwise comparisons with the other three groups. Notably, submucosal clusters 1, 4, and 2 showed the greatest gene expression differences when comparing the L4 group with the other groups (Fig 3A). Consequently, these clusters were chosen for further analysis of specific gene expression differences (Fig 3 and S2–S4 Tables).

Comparisons of gene expression within submucosal clusters 1, 4, and 2 revealed 867, 510, and 880 unique DEGs between L4 and the other groups. Many *KRT* and *SPRR* genes were upregulated across pairwise comparisons in one or more of the three submucosal clusters. Among *KRT* and *SPRR* genes, the L4 versus L1 comparison showed the highest numbers of significantly upregulated genes in clusters 1, 4, and 2. Likewise, seven S100 family genes were upregulated, with *S100A7* and *S100A8* being the most consistently detected across the pairwise group comparisons and submucosal gene clusters. Additional immune-related genes included *CSTA, SLPI, MIF, FABP5,* and *LY6D,* were significantly upregulated in clusters 1, 4, and 2 in the L4 versus L1 comparison (Fig 3 and S2–S4 Tables).

Downregulated DEGs associated with the L4 group encompassed those involved in ECM adhesion and composition (e.g., *SPARCL1*, *PRSS12/23, IGF1, FBLN1),* as well as several collagen-related genes (*COL3A1, COL14A1, COL15A1*, *COL27A1*) across group comparisons. Notably, several genes involved in TGF-β signaling were downregulated in the L4 group. Additional downregulated genes included *ITGB1/5/8, MMP11* and *PTN,* involved in cell adhesion and ECM remodeling [29–31]. Most of these genes showed significance across all three submucosal clusters (1, 4, and 2). Notably, the immunoglobulin transcription factor *TCF4* was also consistently downregulated across all comparisons and clusters. A few immune-related genes such as *FOXO3, IF16, IGLC2,* and *HLA-C* were also downregulated (Fig 3 and S2–S4 Tables).

Collectively, the L4 group had the most pronounced DEGs compared with the other microbiome groups. As also noted for the epithelium, L1 was associated with upregulation of various innate immune and barrier function genes (e.g., *KLK, KRT, SPRRR,* and *S100*) that maintain a protective and balanced immune environment [32].

## Downregulated antibody gene expression in the *L. iners* vs *L. crispatus* -dominated microbiome groups

Downregulation of antibody genes, particularly in Cluster 1 and the Basal layer, in the *L. iners* (L2) compared to the *L. crispatus* (L1) group, stood out when looking at the remaining pairwise group comparisons (S5–S7 Tables). Across all epithelial layers, the comparison displayed downregulation of genes encoding several antibody isoforms, light chains, and the *JCHAIN* gene. A similar pattern was observed within submucosal clusters 1, 4, and 2, although only *IGKC* and *IGLC2* were shared across all three clusters. The most pronounced general downregulation was observed within Cluster 1, located closest to the basal membrane of the epithelium. Finally, decreased expression of *CD81,* involved in B cell receptor signaling was observed in the Upper Intermediate and Basal layers, as well as in all three submucosal clusters when comparing L2 and L1 (Fig 4A and S5 Table).

## Distinct immunoglobulin subclass gene expression profiles across the microbiome groups

Next, pairwise comparisons of gene expression patterns representing the IgG and IgA antibody subclasses revealed distinct, subclass-specific profiles (Fig 4B and S5–S7 Tables). IgA1 and IgA2 gene expression was upregulated in the L1 group, particularly in comparison with L2, across the epithelium and upper submucosal cluster. IgG1 gene expression was elevated in both the L1 and L4 groups. IgG4 was particularly upregulated in the L4 group compared to L2 in the epithelial layers, and against all groups in submucosal Cluster 1. IgG2 exhibited only sporadic differences, and IgG3 showed no significant differences between microbiome groups.

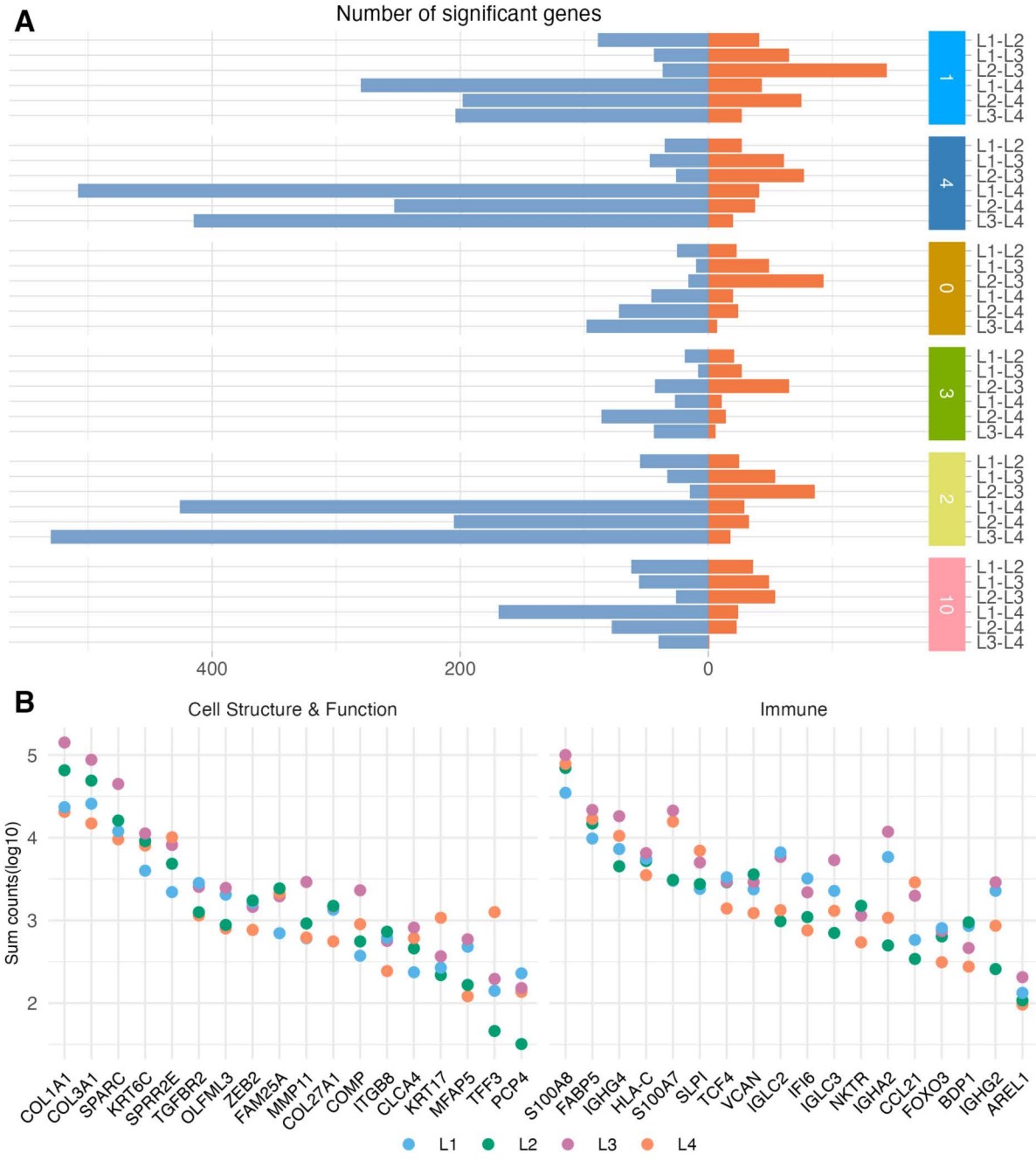

**Fig 3. Differentially expressed genes across submucosal clusters. A** Stacked bar plot of all submucosal DEGs for the Wilcox pairwise model. The x-axis shows number of significantly differentially expressed genes, with orange indicating upregulated and blue downregulated genes. The y-axis indicates the group comparison and the cluster. **B** Dot plot showing the $\log_{10}$ counts of selected genes, grouped by microbiome profiles (L1-L4) and summarized across the submucosal clusters 1, 4 and 2 on the y-axis.

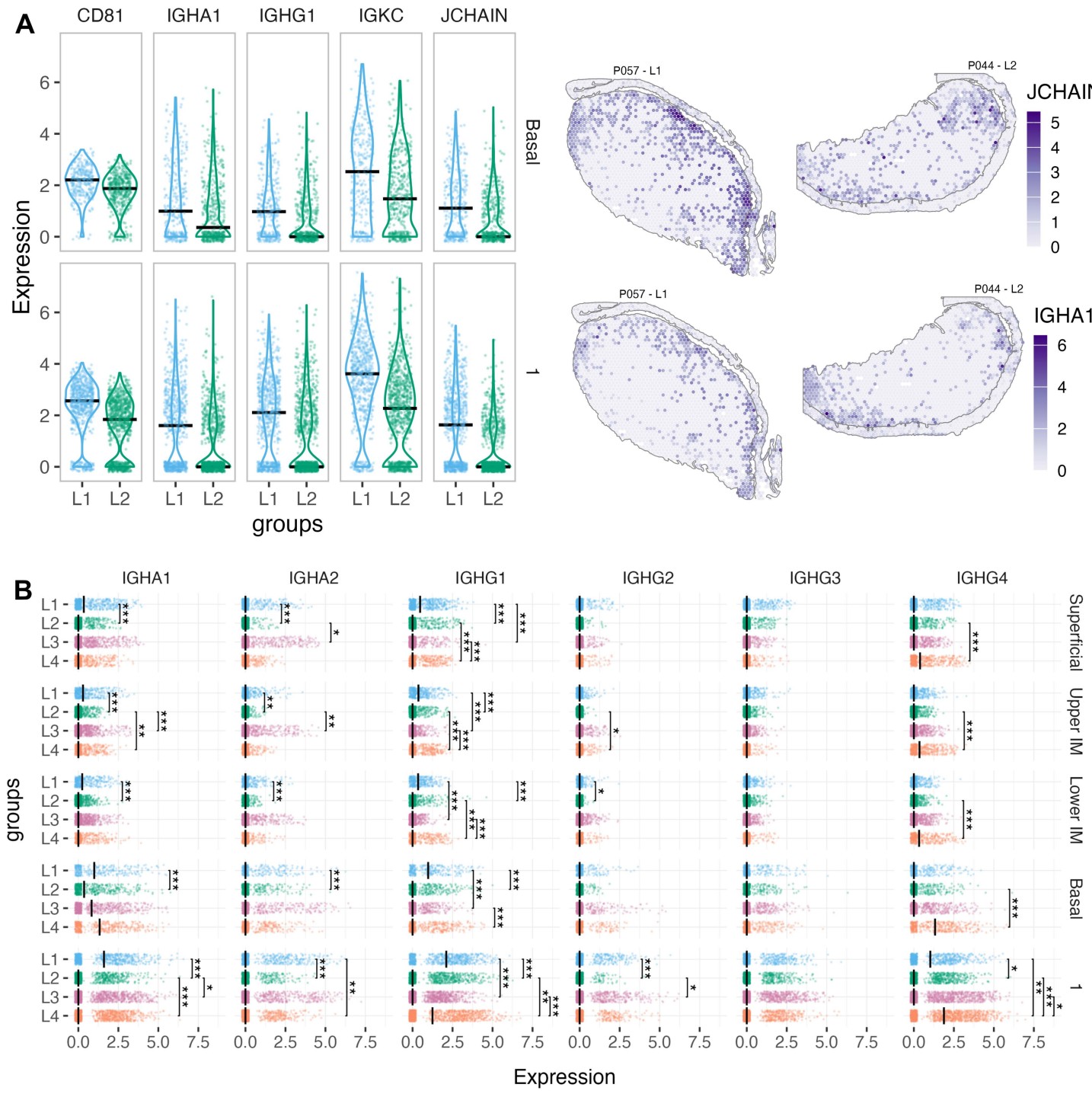

**Fig 4. Antibody gene expression differences between L1 and L2. A** (Left) Violin plots showing the expression levels of CD81, IGHA1, IGHG1, IGKC, and JCHAIN within the Basal layer and Cluster 1. (Right) Spatial expression maps of IGHA2 and IGHA1 on tissue sections from sample P057 (L1) and sample P044 (L2), respectively. Color intensity reflects gene expression levels. **B** Dot plots showing the expression of immunoglobulin subclasses, stratified by microbial groups (L1-L4) and spatial clusters (Superficial - Cluster 1). Statistical significances are indicated as follows: p < 0.05 (*), p < 0.01 (**), and p < 0.001 (***) based on Wilcoxon pairwise comparisons with FDR-adjusted p-values.

## Gene modules defined by spatial transcriptomics reveal inconsistent co-expression patterns between the study groups

To understand the higher-level function of the DEGs, we applied a spatial transcriptomics-optimized WGCNA to the 3,771 unique DEGs identified by all three differential expression methods. This analysis revealed four spatial modules (SMs) representing different biological processes: SM1 (593 genes), SM2 (165 genes), SM3 (715 genes), and SM4 (116 genes) (Fig 5).

SM1 (summarized as 'epithelial barrier function') contained DEGs mostly expressed in the Upper Intermediate layer, with expression in the L2 (*L. Iners) and* L3 (*Gardnerella*) group. The top five hub genes (*PPL, RHCG, KRT13, GJB2, S100A14*) were all related to epithelial barrier function. Gene Ontology (GO) ('keratinocyte differentiation') and Kyoto Encyclopedia of Genes and Genomes (KEGG) ('tight junction') enrichment analysis further confirmed barrier maintenance as a main function related to SM1. However, innate immunity was also represented among the top terms, including 'neutrophil degranulation' (GO) and 'PPAR signaling' (KEGG).

SM2 ('protein production') contained DEGs that were primarily active in the Basal layer, but also in clusters 1 and 4, and exhibited slightly increased expression in the L3 group. Hub genes (*RPS26, RPL18A, RPS21, FAU, RPS15A*) and top GO and KEGG terms indicated protein production ('translation', 'rRNA metabolic process', 'ribosome'). Additional terms suggested increased metabolic and antigen-processing functions.

SM3 genes ('inflammation and processes occurring in the extracellular space') were most predominantly expressed in the L1 and L3 groups and coupled to clusters 2, 9, and 4. The hub genes (*SPARCL1, AEBP1, LGALS1, COL6A2, DCN*), GO terms (e.g., 'extracellular matrix organization'), KEGG terms ('ECM-receptor interaction', 'TGF-β signaling pathway'), and transcription factor analysis indicated processes occurring in the extracellular space. As metallopeptidases are important regulators of inflammation, this module is likely linked to inflammation, which is also suggested by other KEGG terms.

SM4 genes ('regulation of intracellular protein transport and RNA processing') were most strongly expressed in clusters 2 and 0 and were primarily upregulated in the L1 and L2 groups, while L3 and L4 exhibited low SM4 module expression. The hub genes (*PNISR, TTCR, FTX, XIST, ATRX*) have been linked to epigenetic regulation, particularly DNA methylation. Significant GO terms included 'regulation of intracellular protein transport' and 'RNA processing'. Transcription factor analysis revealed activation of key regulatory pathways, including retinoic acid signaling, which influences epithelial differentiation and immune modulation; NF-κB signaling, a central pathway in inflammation and immune responses; and sex hormone signaling, which can modulate mucosal immunity and epithelial cell function.

The four gene modules identified by spatial WGCNA thus represent distinct biological programs that associate with specific spatial and expression patterns across L1 to L4. The data are consistent with microbiome-associated altered gene expression representing structural and immune-related mechanisms throughout the mucosal tissue.

## Correlations of spatial modules with individual bacterial taxa and clinical data modules

The relative abundance of individual bacterial taxa was available from the cervicovaginal microbiome data set and was studied to complement the microbiome group comparisons. The WGCNA-defined gene modules were thereby correlated with the relative abundance of the 16 most common taxa across all clusters (Fig 6). The significant (false discovery rate <0.05) rho values varied from 0.6 to 0.8, and all except for one was for module SM4, which primarily represents the submucosa. Specifically, the *L. crispatus*/*acidophilus* and *L. reuteri*/*oris*/*frumenti*/*antri* taxa correlated positively with SM4, across submucosal clusters 1, 4, 0, and 10 (S5A Fig). According to the SM4 definition, this finding suggests active transcriptional programming as well as retinoic acid, NF-kB, and sex hormone signaling. In contrast, *Gardnerella*, *Atopobium* (now called *Fannyhessya*)*,* and *Dialister* correlated negatively with these functions across the Basal layer, clusters 1, and cluster 4. Next, possible confounding effects between spatial modules and clinical variables were analyzed (S5B Fig). In the Upper Intermediate layer, estradiol correlated positively with SM3, while age and sex work did not show significant

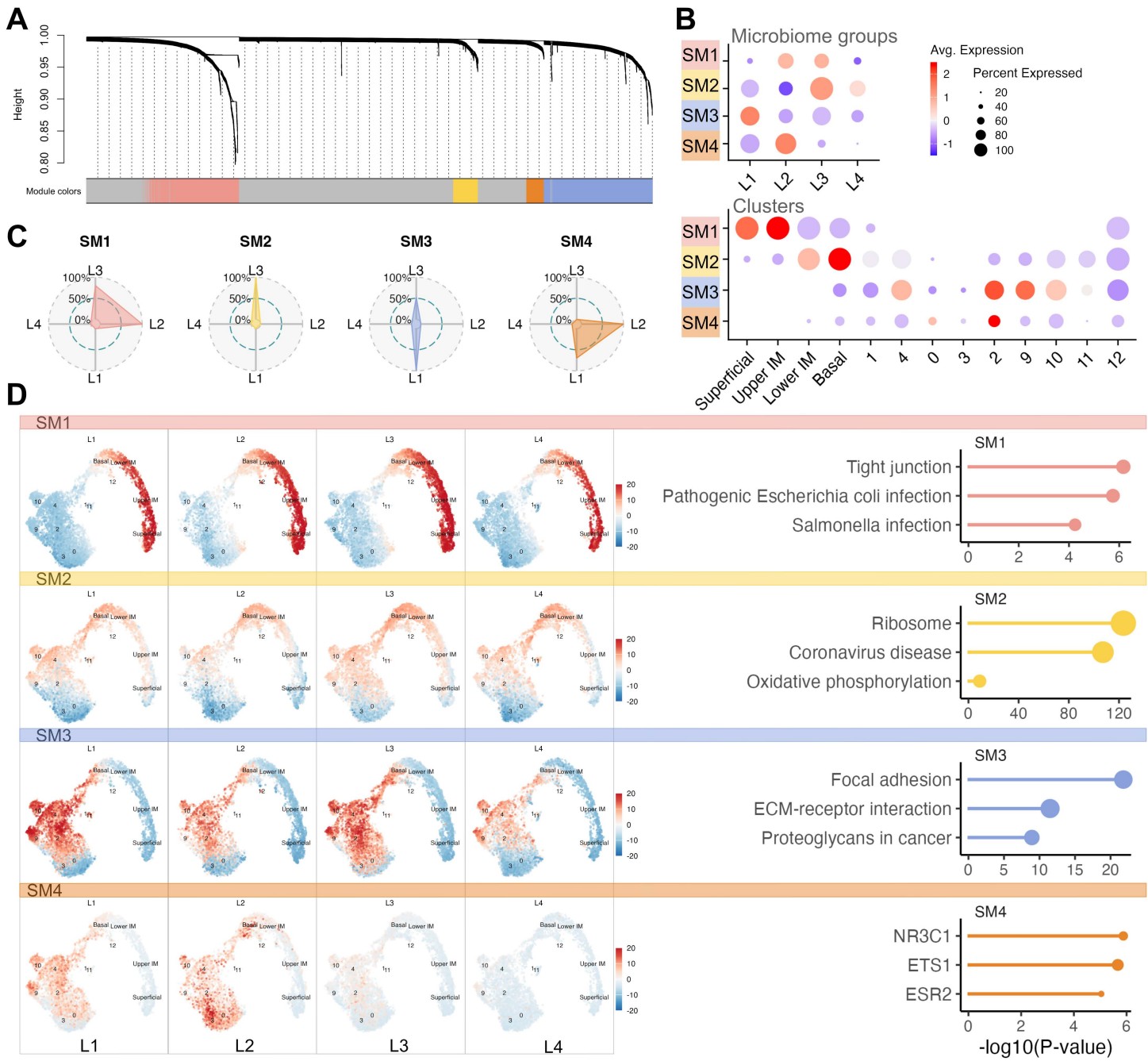

**Fig 5. Weighted Gene Co-expression Network Analysis (WGCNA). A** The dendrogram shows hierarchical clustering of genes based on topological overlap, with each leaf on the dendrogram representing a single gene, and the color at the bottom indicates the co-expression module assignment (SM2 yellow, SM4 orange, SM1 salmon, and SM3 in blue). Gray indicates genes not assigned to any specific module. **B** Dot plot showing relative expression level of each module across microbiome groups and clusters. **C** Radar plot representing microbiome group contribution for each of the modules. **C** Uniform Mani-fold Approximation and Projection (UMAP) showing the module eigengenes (MEs) of each of the four modules across microbiome groups (left), and Enrichment terms connected to each module (right). The enrichment plot has $-\log_{10}$p-value on the x-axis and terms on the y-axis.

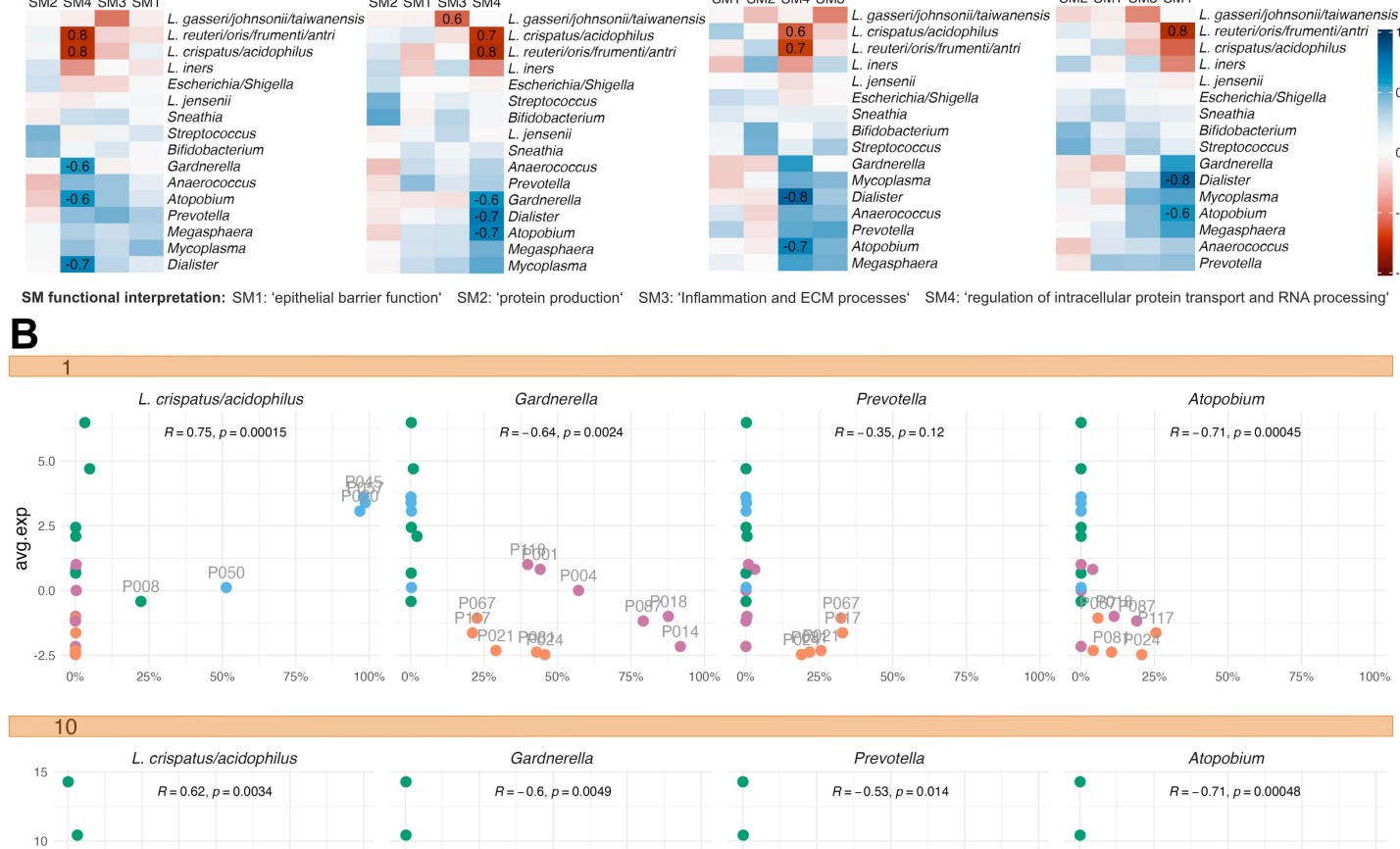

**Fig 6. Associations between microbial taxa and host gene expression modules. A** Heatmaps displaying Spearman's correlation between gene modules and 16 of the most abundant taxa for selected clusters. The numbers indicate significant rho values after FDR adjustment. **B** Dot plots showing the average value of SM4 for individual samples on the y-axis and the relative abundance of specific taxa on the x-axis for Cluster 1 (top) and Cluster 10 (bottom).

correlations with any module in any cluster. The Nugent score, a diagnostic scoring system based on bacterial morphotypes in vaginal smears used to assess bacterial vaginosis, was closely related to the microbiome groups as also evident in the analysis.

In summary, the individual bacterial taxa data were largely in line with data from the study group comparisons, thus supporting the comparisons for the *L. crispatus*, *Gardnerella*, and highly diverse groups, as described above.

## Discussion

We demonstrated that the cervicovaginal microbiome correlates with various host transcriptomic patterns in ectocervical mucosal tissue samples from Kenyan sex workers. By mapping host mRNA expression at near single-cell resolution in the tissue, we were able to identify the spatial localization of these correlates. The highly diverse microbiome group was associated with the most pronounced differences in gene expression, primarily located around the epithelial membrane, compared to microbiome groups dominated by *L. crispatus, L. iners,* and *Gardnerella* taxa. These differences are theoretically related to disruption of epithelial and ECM structures and to augmented immune responses. While the *L. crispatus*-dominated cervicovaginal microbiome is often contrasted with the pro-inflammatory profile of highly diverse microbial communities, this state may not be immunologically quiescent. Rather, the transcriptional signatures implied activities including tight junction maintenance, antimicrobial peptide production, and immune tolerance. This protective status gradually shifted toward other patterns of immune activation and epithelial remodeling in the *L. iners* and *Gardnerella* groups. This finding supported previous studies showing that *L. crispatus* interacts with anti-inflammatory innate immune receptors, while *L. iners* and *Gardnerella* interact with both pro- and anti-inflammatory receptors [33].

The gene expression profiles observed across all study groups were not restricted to the apical epithelial surface, which is in direct contact with the cervicovaginal microbiome. The apical layer is structurally more permeable than the deeper layers because it lacks classical cell-cell adhesions and can thereby allow passage of macromolecules such as IgG [34]. Thus, even though this layer may harbor bacteria, it is assumed that deeper layers are largely bacteria-free under non-inflammatory conditions. It is therefore interesting that our data indicated microbiome-associated gene expression differences representing activated host responses throughout the entire ectocervical mucosa.

A shift from an *L. crispatus*-dominated microbiome toward a more highly diverse microbiome can be associated with sexual health and reproductive disorders, although the pathogenicity of these shifts varies across populations [35–37]. In the present study, among nearly 26,000 genes, approximately 12% varied across the four microbiome-based study groups, which may contribute to the pathogenesis associated with the cervicovaginal microbiome. In the highly diverse study group, the most predominant gene expression differences represented dysregulated functions, including epithelial barrier maintenance, ECM structure, and immune responses, the latter also including TGF-β regulation. These differences were primarily identified in the lower part of the epithelium and upper part of the submucosa, where cells are densely clustered.

Another mechanism associated with the highly diverse microbiome profile included keratin regulation. Keratins are differentially expressed across the epithelium to support functional keratinocyte differentiation and maintain cellular stability and resilience. For example, compared with the *Gardnerella*-dominated group, the highly diverse group showed significantly altered, spatially restricted regulation of several keratins across the epithelium. Dysregulation of several of these specific keratins is seen in inflammatory skin conditions such as psoriasis and atopic dermatitis, implicating their involvement in a pathological mucosal response [38]. Other mechanisms associated with the highly diverse microbiome group included the *KLK* and *SPRR* gene families. These differences were spatially restricted to different layers of the epithelium. *KLK* genes encode serine proteases that are abundantly expressed in the genital mucosa, playing a crucial role in physiological desquamation of epithelial cells and activation of antimicrobial proteins [38]. Upregulation of *KLK* genes indicates an inflammatory response, along with significant upregulation of counteracting protease inhibitors. Increased *KLK* activity has previously been associated with a disturbed vaginal microbiome [39]. *SPRR*s are antimicrobial proteins that are part of the mucosal response to microbial presence [40]. We demonstrated broad microbiome-associated effects on several *SPRR* genes across the mucosa. Among these mechanistic alterations, the superficial epithelial layer demonstrated the most pronounced induction of *KLK* and *SPRR* genes, associating inflammation with the superficially located, highly diverse microbiome.

The highly diverse microbiome group was also associated with distinct patterns of immunoglobulin subclass gene expression, including an upregulation of IgG1 and IgG4 across the epithelium and upper basal layer. This contrasted to the *L. crispatus*-dominated group, which was mainly associated with upregulation of IgA1 and IgA2 in the same mucosal

regions. At mucosal surfaces, IgG1 typically mediates pathogen clearance through inflammation and opsonization, IgG4 is associated with immune regulation and tolerance, IgA1 neutralizes pathogens without triggering inflammation and prevents adhesion of microbes to epithelial cells, while IgA2 has a more pro-inflammatory antimicrobial effect. A more detailed understanding of host-microbiota interactions in the hormone-regulated vaginal mucosa is however limited and largely derived from observational and cross-sectional studies, which complicates causality issues [41–43]. Notably, mucosal levels of pro-inflammatory cytokines have been correlated with increased total genital mucosal antibody levels [44] and reduced IgA coating of bacteria [45]. The mechanisms by which microbe-binding antibodies modulate host immune responses or shape vaginal bacterial colonization remain unclear but may involve enhanced IgA coating of bacteria in women with *L. crispatus*-dominated vaginal microbiota [46] and IgA-mediated reduction of total bacterial abundance [47].

To further define the function of the DEGs associated with the study groups, a gene co-expression network analysis revealed four gene modules, each representing distinct functions. The analysis demonstrated that the *Gardnerella*-dominated microbiome group was also linked to compromised epithelial integrity (SM1) and increased metabolic and antigen-processing functions (SM2). The spatial analysis localized these functions to the proximity of the basal membrane. Both the *L. crispatus* and *Gardnerella* groups exhibited activity in relation to submucosal tissue remodeling and immune activity (SM3). In addition, both the *L. crispatus* and *L. iners* groups were associated with active transcriptional activity and hormonal responsiveness (SM4). The correlations of individual taxa in relation to the spatial gene modules were in line with the results achieved from the study group analyses. However, these analyses were restricted as the taxa did not always represent individual species and did not cover intra-species variations. Such variations can have a large impact on immune functions [48,49].

Our data on gene expression differences across the epithelium and submucosa indicate several mechanisms through which the cervicovaginal microbiome could influence susceptibility to HIV. However, the causal direction between microbiome composition and host gene expression cannot be determined by the present cross-sectional study design. We cannot exclude the possibility that host transcriptional states influence microbiome structure rather than vice versa. Nevertheless, the most prominent finding was that transcriptional differences extended across the tissue, from the superficial epithelial layer to the deep submucosal area, the latter defined by smooth muscle marker genes, with the most pronounced changes observed adjacent to the basal membrane. These results are in line with previously presented findings, including microbiome-associated alterations of immune-related mechanisms that can facilitate HIV penetration [8,50,51] and trigger recruitment of activated CD4 + T cells to the genital mucosa, promoting HIV spread [8,52,53]. We find it important to study how the cervicovaginal microbiome relates to the cervical mucosa in female sex workers from sub-Saharan Africa because they are a key target population for ongoing efforts to reduce global HIV transmission [54]. Additionally, it is particularly significant to conduct microbiome studies across continents, as the microbiome and its implications for sexual and reproductive health vary globally [7,55]. For example, some cohorts of sex workers have been shown to have a 21 times greater risk of HIV infection than women in the general population [56].

While these findings offer valuable insights, several limitations must be acknowledged. First, as already mentioned, this study is observational in nature, and we cannot infer directionality in the microbial–host transcriptional associations. Second, the 10x Genomics Visium platform has a 55 μm spot size, which precludes single-cell resolution and introduces gaps between spots. As a result, spatial specificity should be interpreted at a regional rather than single-cell level. Nevertheless, the consistent layer- and cluster-specific transcriptional patterns observed across comparisons support the robustness of our findings despite this resolution constraint. This resolution also limited our ability to study immune cell populations in detail, highlighting the need for higher-resolution or integration with single-cell spatial technologies. Third, transcriptomic data reflect mRNA abundance, which may not always correlate directly with protein expression. Therefore, validation of key findings at the protein level (e.g., immunohistochemistry or proteomic analyses) is necessary to confirm biological relevance. Fourth, while sex workers represent a key population for HIV research, their unique immune environment, shaped by frequent

exposure to factors such as high partner turnover, increased risk of sexually transmitted infections, and repeated mucosal inflammation, may limit the generalizability of our findings to non–sex worker populations. Similarly, because microbiome composition varies across ethnic groups, our cohort of African women may not fully represent microbiome–host interactions in other populations. Finally, spatial transcriptomics does not capture dynamic temporal changes or cell–cell interactions in real time, which may be critical in understanding mucosal immune responses and microbiome dynamics.

Looking ahead, a deeper understanding of the mechanisms that govern host–microbiome interactions at spatial resolution within the female genital tract mucosa will be important. Recent technological advances, including high-resolution imaging and integrative multi-omics, together with an improved understanding of mucosal immune responses, provide renewed opportunities to dissect these interactions. Harnessing these approaches will be critical for defining the immunological landscape of the female genital tract and for informing strategies to reduce HIV susceptibility.

## Methods

### Ethical statement

This study received approval from the ethical review boards of the University of Manitoba, University of Nairobi's Kenyatta National Hospital, and regional Ethical Review Board in Stockholm. Written informed consent was obtained from all participants.

### Study participants and clinical samples

All clinical samples were collected as part of a nested study using the Longitudinal Assessment of Immune Quiescence within a cohort of Kenyan sex workers [22,57]. Nine spatial transcriptomics samples from the same cohort were selected from our published data [13,14] to complement the 12 newly sequenced samples. The inclusion criteria for the larger study included individuals aged 18–50 years who were not pregnant or breastfeeding, not menopausal, had no history of hysterectomy, and tested negative for *Treponema pallidum, Neisseria gonorrhoeae,* and *Chlamydia trachomatis.*

Briefly, tissue samples up to 3 mm were obtained by a senior gynecologist. The samples were immediately snap frozen and stored at −80°C. Participants agreed to abstain from sexual activity for 2 weeks after sampling for safety reasons. Adherence was supported through text reminders, on-site prostate-specific antigen detection, and financial compensation [57]. A clinical examination 3–5 days post-biopsy assessed healing and confirmed compliance with instructions.

### Sample processing and Visium library preparation

Frozen tissue sections were cut at a thickness of 10 µm using a cryostat maintained at a low temperature, mounted onto Visium Spatial Gene Expression slides (10x Genomics), and carefully positioned within the 6.5 mm$^2$ oligonucleotide-barcoded capture regions. Following tissue fixation, standard hematoxylin and eosin staining was performed directly on the slides. High-resolution imaging was conducted prior to proceeding with library construction, which followed the Visium Spatial Gene Expression protocol provided by 10x Genomics.

Sequencing was carried out on the NovaSeq 6000 platform (NovaSeq Control Software v1.8.0, RTA v3.4.4) using an S4 flow cell in 'NovaSeqXp' workflow mode. The run configuration included paired-end reads with the following structure: 151 nucleotides for Read 1, 19 nucleotides for Index 1, 10 nucleotides for Index 2, and 151 nucleotides for Read 2. Raw base call (BCL) files were converted to FASTQ format using bcl2fastq (v2.20.0.422) from the CASAVA software suite. All sequencing data were encoded using the Sanger/Phred+33/Illumina 1.8 + quality scale.

### Data preprocessing

The samples were selected from three separate sequencing runs. The preprocessing details of individual runs can be found in the Gene Expression Omnibus (GEO) public repository (GSE217237). In short, the raw reads were processed using the spaceranger tool (v1.2.0/v2.0.1/v2.1.0, 10x Genomics) and aligned to the human reference genome (*Homo*

*sapiens*, GRCh38-2020-A). Gene expression matrices and tissue images for each sample were then imported as Seurat objects using the load10x_spatial and Read10X_Image functions of the Seurat package [58].

Highly variable genes were identified individually for each sample using the FindVariableFeatures function with the 'vst' method and a target of 2,000 genes. To ensure cross-sample relevance, only highly variable genes detected in at least two samples were retained, with immunoglobulin and T-cell receptor (VDJ) genes excluded from further analysis. The resulting gene set was normalized using the NormalizeData function with log-normalization, followed by scaling via ScaleData.

Principal component analysis (PCA) was then applied to this processed gene set using the RunPCA function. The PCA results served as the input for sample integration using Harmony, specifying the RNA assay and using the top 50 principal components. For downstream clustering, Harmony-corrected dimensions 1 through 30 were used to construct a k-nearest neighbors graph (FindNeighbors), followed by community detection using the Louvain algorithm (FindClusters) with k.param = 15 and a resolution of 0.7. To visualize the data, two-dimensional UMAP embedding was performed using the top 50 Harmony components (RunUMAP).

Tissue regions corresponding to epithelial and submucosal compartments were manually annotated by outlining the mucosal boundary. These hand-drawn polygons were used to assign spatial transcriptomic spots, with partially overlapping spots manually curated to ensure accurate labeling.

### Differential gene expression analysis

Differentially expressed gene (DEG) analysis was performed using the FindAllMarkers function (test.use = Wilcox) of the Seurat package to identify marker genes for each cluster. To compare DEGs between study groups, we applied both pseudo-bulk analysis and the Wilcox test (S1 Text). For pseudo-bulk analysis, spots from all samples were pooled by regions defined by clustering, and standard DEG methods were used with the quasi-likelihood model of the edgeR package [59]. Two models were considered: one across all groups and another for individual group comparisons. The Wilcox test was applied to a reduced dataset, created by randomly sampling 25 spots from each epithelial cluster and 50 spots from each submucosal cluster per sample. For the Basal layer, which contained a mix of submucosal and epithelial spots, 20 spots of each type were included per sample.

### High-dimensional weighted correlation network analysis

Weighted gene co-expression network analysis (WGCNA) was conducted on the DEGs to construct a co-expression network at the spot level using the 'hdWGCNA' package [60]. To reduce the complexity of the spatial data, the 'Metacell-sByGroups' function was used to create a metacell gene expression matrix for network construction. For the analysis, an appropriate soft threshold power of 9 was determined using the 'TestSoftPowers' function. All analyses were performed following the standard procedures outlined in the hdWGCNA tutorial.

### Supporting information

**S1 Fig. 16S rRNA gene–based proportions and community structure of vaginal taxa. A** Bar plot of 16S taxonomical relative abundance for each sample, ordered by microbial groups L1-L4. **B** Non-metric MultiDimensional Scaling (NMDS) of normalized taxonomic counts. Ellipses encircle the samples belonging to the four microbiome groups L1 (blue), L2 (green), L3 (pink) and L4 (orange) respectively.
(TIFF)

**S2 Fig. Quality control.** Violin plots of total transcripts/counts **A** and genes/features **B** for individual samples. The study groups exhibited the following distribution of total counts/transcripts: (median: L1: 2,176; L2: 2,884; L3: 3,226; L4: 1,422) and genes/features (median: L1: 1,237; L2: 1,438; L3: 1,601; L4: 863).
(TIFF)

**S3 Fig. Clustering.** Spots colored by unsupervised Louvain clustering.
(TIFF)

**S4 Fig. Differential gene expression in epithelial and submucosal clusters.** Volcano plots of all **A** epithelial and **B** submucosal DEGs for the Wilcox pairwise model, for each cluster on the x-axis and log fold change on the y-axis.
(TIFF)

**S5 Fig. Correlation of host gene expression modules with microbial taxa and clinical variables.** Heatmaps displaying Spearman's correlation between gene modules and **A** 16 of the most abundant taxa for all clusters or **B** clinical variables of interest. The numbers indicate significant rho values after FDR adjustment.
(TIFF)

**S1 Table. One vs rest differential gene expression.** The top marker genes for each cluster, identified through one-vs-rest differential expression analysis The list of genes in each cluster is shown in separate sheets.
(XLSX)

**S2 Table. Differential gene expression between L1 and L4.** Results of the differential gene expression analysis between the microbiome groups '*L. crispatus*' (L1) and 'highly diverse' (L4). Sheets 1 and 2 contain the top 15 most upregulated and downregulated genes, ranked by average $\log_2$ fold change, for epithelial clusters (5–8) and submucosal clusters (0–4, 9, and 10), respectively. Sheet 3 provides a summary of the number of significant DEGs (adjusted p-value < 0.05) across clusters, including counts of up- and downregulated genes. The remaining sheets contain the full lists of differentially expressed genes for each cluster.
(XLSX)

**S3 Table. Differential gene expression between L2 and L4.** Result of the differential gene expression analysis between the microbiome group defined as '*L. iners*' (L2) and 'highly diverse' (L4). Sheets 1 and 2 contain the top 15 most upregulated and downregulated genes, ranked by average $\log_2$ fold change, for epithelial clusters (5–8) and submucosal clusters (0–4, 9, and 10), respectively. Sheet 3 provides a summary of the number of significant DEGs (adjusted p-value < 0.05) across clusters, including counts of up- and downregulated genes. The remaining sheets contain the full lists of differentially expressed genes for each cluster.
(XLSX)

**S4 Table. Differential gene expression between L3 and L4.** Result of the differential gene expression analysis between the microbiome group defined as '*Gardnerella*' (L3) and 'highly diverse' (L4). Sheets 1 and 2 contain the top 15 most upregulated and downregulated genes, ranked by average $\log_2$ fold change, for epithelial clusters (5–8) and submucosal clusters (0–4, 9, and 10), respectively. Sheet 3 provides a summary of the number of significant DEGs (adjusted p-value < 0.05) across clusters, including counts of up- and downregulated genes. The remaining sheets contain the full lists of differentially expressed genes for each cluster.
(XLSX)

**S5 Table. Differential gene expression between L1 and L2.** Result of the differential gene expression analysis between the microbiome group defined as '*L. crispatus*' (L1) and '*L. iners*' (L2). Sheets 1 and 2 contain the top 15 most upregulated and downregulated genes, ranked by average $\log_2$ fold change, for epithelial clusters (5–8) and submucosal clusters (0–4, 9, and 10), respectively. Sheet 3 provides a summary of the number of significant DEGs (adjusted p-value < 0.05) across clusters, including counts of up- and downregulated genes. The remaining sheets contain the full lists of differentially expressed genes for each cluster.
(XLSX)

**S6 Table. Differential gene expression between L1 and L3.** Result of the differential gene expression analysis between the microbiome group defined as '*L. crispatus*' (L1) and '*Gardnerella*' (L3). Sheets 1 and 2 contain the top 15 most upregulated and downregulated genes, ranked by average $\log_2$ fold change, for epithelial clusters (5–8) and sub-mucosal clusters (0–4, 9, and 10), respectively. Sheet 3 provides a summary of the number of significant DEGs (adjusted p-value < 0.05) across clusters, including counts of up- and downregulated genes. The remaining sheets contain the full lists of differentially expressed genes for each cluster.
(XLSX)

**S7 Table. Differential gene expression between L2 and L3.** Result of the differential gene expression analysis between the microbiome group defined as '*L. iners*' (L2) and '*Gardnerella*' (L3). Sheets 1 and 2 contain the top 15 most upregulated and downregulated genes, ranked by average $\log_2$ fold change, for epithelial clusters (5–8) and submu-cosal clusters (0–4, 9, and 10), respectively. Sheet 3 provides a summary of the number of significant DEGs (adjusted p-value < 0.05) across clusters, including counts of up- and downregulated genes. The remaining sheets contain the full lists of differentially expressed genes for each cluster.
(XLSX)

**S1 Text. More detailed descriptions of the differential gene expression workflows are provided, including data preprocessing, normalization, pseudo-bulk aggregation, and the statistical frameworks used for both global (across groups) and pairwise comparisons.** The rationale for combining pseudo-bulk and spot-level (Wilcoxon) approaches, as well as the steps taken to minimize biases such as pseudo-replication and inflated p-values, are explained in detail. Additional information is also provided on quality control, filtering thresholds, and the criteria used to define significantly differentially expressed genes (DEGs).
(PDF)

## Acknowledgments

The authors extend their gratitude to the research participants, Professor Julius Oyugi, and the Majengo clinical staff for their valuable support and time. Special thanks to Dr. Angela Muliro for conducting the clinical examinations and sample collections, Juliana Cheruiyot for her coordination of the research participants. The authors further acknowledge support from the National Genomics Infrastructure in Stockholm funded by the Science for Life Laboratory, the NAISS/Uppsala Multidisciplinary Center for Advanced Computational Science for assistance with massively parallel sequencing and access to the UPPMAX computational infrastructure.

## Author contributions

**Conceptualization:** Vilde Kaldhusdal, Joshua Kimani, Keith R. Fowke, Douglas S. Kwon, Kristina Broliden.

**Data curation:** Vilde Kaldhusdal, Julie Lajoie, Gabriella Edfeldt.

**Formal analysis:** Vilde Kaldhusdal.

**Funding acquisition:** Vilde Kaldhusdal, Adam D. Burgener, Keith R. Fowke, Kristina Broliden.

**Investigation:** Vilde Kaldhusdal, Mathias Franzén Boger, Adam D. Burgener, Julie Lajoie, Kenneth Omollo, Joshua Kimani, Annelie Tjernlund, Gabriella Edfeldt, Kristina Broliden.

**Methodology:** Vilde Kaldhusdal, Mathias Franzén Boger, Adam D. Burgener, Julie Lajoie, Kenneth Omollo, Joshua Kimani, Annelie Tjernlund, Gabriella Edfeldt, Kristina Broliden.

**Project administration:** Joshua Kimani, Keith R. Fowke, Douglas S. Kwon, Kristina Broliden.

**Resources:** Joshua Kimani, Keith R. Fowke, Douglas S. Kwon, Kristina Broliden.

**Software:** Vilde Kaldhusdal, Gabriella Edfeldt.

**Supervision:** Adam D. Burgener, Joshua Kimani, Annelie Tjernlund, Keith R. Fowke, Douglas S. Kwon, Kristina Broliden.

**Validation:** Vilde Kaldhusdal, Mathias Franzén Boger, Julie Lajoie, Douglas S. Kwon, Gabriella Edfeldt, Kristina Broliden.

**Visualization:** Vilde Kaldhusdal.

**Writing – original draft:** Vilde Kaldhusdal, Mathias Franzén Boger, Kristina Broliden.

**Writing – review & editing:** Vilde Kaldhusdal, Mathias Franzén Boger, Adam D. Burgener, Julie Lajoie, Kenneth Omollo, Joshua Kimani, Annelie Tjernlund, Keith R. Fowke, Douglas S. Kwon, Gabriella Edfeldt, Kristina Broliden.

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
