## [Decision Letter · Decision Letter 0]

19 Aug 2025

The cervicovaginal microbiome impacts spatially restricted host transcriptional signatures throughout the human ectocervical epithelium and submucosa

PLOS Pathogens

Dear Dr. Kaldhusdal,

Thank you for submitting your manuscript to PLOS Pathogens. After careful consideration, we feel that it has merit but does not fully meet PLOS Pathogens's publication criteria as it currently stands. Therefore, we invite you to submit a revised version of the manuscript that addresses the points raised during the review process.

Please submit your revised manuscript within 60 days Oct 18 2025 11:59PM. If you will need more time than this to complete your revisions, please reply to this message or contact the journal office at plospathogens@plos.org. Please include the following items when submitting your revised manuscript:

We look forward to receiving your revised manuscript.

Kind regards,

Julia Oh

Academic Editor

PLOS Pathogens

Michael Otto

Section Editor

PLOS Pathogens

Editor-in-Chief

PLOS Pathogens

orcid.org/0000-0003-2946-9497

Michael Malim

PLOS Pathogens

orcid.org/0000-0002-7699-2064

**Journal Requirements:**

At this stage, the following Authors/Authors require contributions: Vilde Kaldhusdal. Please ensure that the full contributions of each author are acknowledged in the "Add/Edit/Remove Authors" section of our submission form.

4) We notice that your supplementary Figures are included in the manuscript file. Please remove them and upload them with the file type 'Supporting Information'. Please ensure that each Supporting Information file has a legend listed in the manuscript after the references list.

Potential Copyright Issues:

i) Figure 1A. Please confirm whether you drew the images / clip-art within the figure panels by hand. If you did not draw the images, please provide (a) a link to the source of the images or icons and their license / terms of use; or (b) written permission from the copyright holder to publish the images or icons under our CC BY 4.0 license. Alternatively, you may replace the images with open source alternatives. See these open source resources you may use to replace images / clip-art:

6) Kindly revise your competing statement to align with the journal's style guidelines: 'The authors declare that there are no competing interests.'

**Reviewers' Comments:**

Reviewer's Responses to Questions

**Part I - Summary**

Reviewer #1: The study presents a spatial transcriptomic analysis of ectocervical tissue from Kenyan female sex workers, categorized by cervicovaginal microbiome composition. It identifies microbiome-associated transcriptional differences across epithelial and submucosal compartments, with implications for mucosal barrier integrity and immune activation. I found the study to be methodologically strong and biologically relevant, with clear potential for advancing our understanding of host-microbiome interactions in the context of HIV susceptibility. My review focuses on improving clarity, figure annotation, and narrative structure, and emphasizes the need to avoid causal language in interpreting cross-sectional data while encouraging deeper integration with established mucosal immunology literature.

Reviewer #2: Kaldhusdal et al present new findings on the host transcriptional profile of the human cervicovaginal mucosa incorporating both spatial information and vaginal microbial composition. While the vaginal microbiota is associated with reproductive health outcomes, mechanisms dictating these associations are still poorly understood. In this manuscript, the authors present a new spatial transcriptomics data set of 21 human ectocervical biopsies and correlate these data with vaginal 16S rRNA gene sequencing, grouped into 4 community types. Some of the main findings include distinct transcriptional profile differences in the high diversity microbial group, concentrating in the lower epithelial layers and submucosa, and decreased expression of antibody related genes in the L. iners dominant group, and consistent spatial module correlations with bacterial taxa. Overall, this manuscript presents new findings of high relevance to the field, and is well-written with appropriate experimental design and data interpretation. Specific comments (below) are minor and meant to improve the clarity of the work.

Reviewer #3: This manuscript by Kaldhusdal and colleagues investigates how vaginal microbiome composition influences gene expression in cervical tissues, using spatial transcriptomics to map changes across distinct mucosal compartments. Cervicovaginal microbiota are known to affect susceptibility to HIV and other STIs; this work aims to define spatially resolved molecular pathways underlying these associations.

Building on earlier bulk transcriptomic studies by the Broliden group in Kenyan sex workers, the authors applied 10x Visium spatial transcriptomics to ectocervical tissue samples from the longitudinal Sex Worker Outreach Program in Pumwani, Nairobi. Twenty-one samples, representing four microbiome community types, were selected based on microbial profile, RNA quality, tissue integrity, and participant characteristics.

Spatial mapping revealed clear compartmentalization: three epithelial layers (Superficial, Upper Intermediate, Lower Intermediate) and a Basal layer at the epithelial–submucosal interface, alongside heterogeneous submucosal clusters enriched for inflammatory, endothelial, fibroblast, and smooth muscle signatures. Epithelial regions showed programs related to keratinocyte differentiation, cornification, barrier function, and immune activity, while submucosa displayed more diverse transcriptional landscapes.

Differential gene expression analyses (across, pairwise, and pseudo-bulk) identified the highly diverse microbiome group (L4) as the most transcriptionally distinct. In the epithelium, L4 exhibited extensive changes in the Lower Intermediate and Basal layers, with >1,000 unique DEGs per pairwise comparison, including upregulation of genes involved in barrier maintenance, repair, antimicrobial defense, and immune responses, and downregulation of barrier, immune regulatory, and metabolic genes. These shifts encompassed keratins, protease inhibitors, integrins, desmosomal proteins, collagens, and regulators such as STAT3 and IL18.

In the submucosa, L4 showed strong upregulation of keratin (KRT), small proline-rich protein (SPRR), S100 family, and immune-related genes, with concurrent downregulation of extracellular matrix, adhesion, and TGF-β signaling components, suggesting heightened immune activation with reduced structural stability. Separately, the L. iners–dominated group (L2) demonstrated broad downregulation of antibody and B cell–related genes compared to L. crispatus (L1), most markedly at the epithelial–submucosal boundary.

Spatial WGCNA defined four gene modules: SM1 (epithelial barrier function), SM2 (protein production), SM3 (extracellular matrix/inflammation), and SM4 (intracellular transport/RNA processing), each with distinct spatial patterns and microbiome associations.

Correlation with individual bacterial taxa revealed strong links between SM4 and submucosal activity: positive with L. crispatus and L. reuteri, negative with Gardnerella, Atopobium, and Dialister. Clinical variables showed minimal associations, and findings aligned closely with microbiome group comparisons, reinforcing distinct host–microbe interaction patterns in L. crispatus-, Gardnerella-, and highly diverse communities.

Overall, this is an innovative and timely study applying spatial transcriptomics to investigate the spatially resolved impact of cervical microbiome composition on mucosal gene expression. The experimental design is strong, the dataset is rich, and the analyses yield novel biological insights into epithelial and submucosal immune programs, barrier function, and host–microbe interactions. These findings extend prior bulk transcriptomic work and provide new mechanistic hypotheses with potential clinical and translational relevance. My overall enthusiasm for the manuscript is high.

**Part II – Major Issues: Key Experiments Required for Acceptance**

Reviewer #1: I do not believe additional experiments are required to validate the study’s conclusions. However, I do recommend substantial revisions to the language used in interpreting the data. The manuscript frequently implies causality between microbiome composition and host gene expression, despite being based on cross-sectional data. To ensure the conclusions are appropriately supported, the authors should revise the discussion and results sections to consistently reflect associative relationships rather than mechanistic ones. This change is essential for maintaining interpretive rigor, especially in a field where microbiome findings often inform clinical or public health strategies.

Reviewer #2: (No Response)

Reviewer #3: 1. First, the 10x Genomics Visium platform’s 55 µm spot size precludes single-cell resolution, with substantial gaps between spots; while this is an intrinsic limitation of the technology at the time of the study, it warrants more explicit discussion in the context of interpreting spatial specificity.

2. The use of unsupervised clustering without mapping to single-cell reference datasets led to mixed-cell type clusters (e.g., containing both B-cell and keratinocyte markers), which could complicate biological interpretation.

3. Finally, the causal direction between microbiome composition and host gene expression remains unresolved; the data cannot exclude the possibility that host transcriptional states influence microbiome structure rather than vice versa. There is no discussion of this fact or suggestions of future directions to elucidate this.

**Part III – Minor Issues: Editorial and Data Presentation Modifications**

Reviewer #1: Introduction:

Line 98: Requires reference. “An optimal cervicovaginal microbiome is typically dominated by Lactobacillus species, which help maintain an acidic environment that protects against genital infections.”

Line 100: Requires reference. “However, an anaerobic, highly diverse microbiome dominated by non-Lactobacillus bacteria is more commonly observed in women in sub-Saharan Africa.”

Lines 118-125: The end of the introduction refers to a prior study as the basis for this manuscript, making the current text inaccessible to naïve readers. I recommend adding 1-2 sentences briefly outlining the studies referenced to set up the study design of this manuscript (cross-sectional or longitudinal).

Results:

It seems that the first half of the results were written by one person (DEG analysis) and WGCNA section was written by someone else. This is evident by the way the figures are referred to (per panel vs. whole figure), as well as how the text is written (the second half is clearer). The following comments are intended to improve the readability and accessibility to the study results.

Line 131: Add space. “The previously published16S rRNA V4 gene sequencing data”

Figure S1: Is there a way to indicate on the figure to which study group (L1-L4) the samples belong?

Lines 147-289: Across this section of results, I broadly recommend altering the topic sentence of each paragraph with the “conclusive” statement currently found at the end of each paragraph. Such a strategy would improve contextualizing the specific results reported. For example, in 276-288, instead of the current topic sentence “We next investigated the remaining pairwise group comparisons (Table S5–7)”, replace this with “We observed tissue-wide downregulation of antibody genes in the L. iners (L2) compared to the L. crispatus (L1) group (Table S5–7)”.

Line 158-159: “Top marker genes” is vague. I recommend updating the sentence to “The top marker genes for each cluster, identified through one-vs-rest differential expression analysis, largely aligned with their spatial location within the tissue architecture”. Also, I think the reference to Fig. 1B should be replaced with “Fig. 1C, top panel”.

Table S1: Please add this to the table caption the comparisons being made resulting in logFC values? For example, for the first gene, CRCT1, what is the comparison resulting in the reported increased expression (logFC=3.24)?

Lines 167-168: I suggest updating this sentence with: “The gene expression profiles of the submucosal clusters were less discernable and displayed a less structured spatial distribution compared to the epithelial layers (Fig. 1C, lower panel; Table S2)”

Figure 1D: What are the units of the average expression indicated in the legend?

Figure 2C: It may be helpful to indicate the layer in which the gene is enriched.

Lines 206-218 & Figure 2C: The text would be better supported by Figure 2C if there were an indication of the epithelial layer (stratify by or annotate with epithelial layer). Also, regarding the IG genes, my examination of the data suggest that the Upper IM actually has a “broader range” of upregulated genes (see figure below).

Lines 220-233: OLFM4 is downregulated by less than 1-fold in each comparison. While the association is strong, I wouldn’t call the level of downregulation strong. Also, the text is a bit scattered after Line 222 (the first sentence). I appreciate that these data are multi-faceted and complex reflecting the nuanced relationships between the host tissue and exposure to the VMB. This is why a well-structured and flowing narrative is especially essential. As is, it is more like a list of findings. For example, KLK and KRT genes are referred to again, though they were mentioned two paragraphs up.

Supplemental Tables. Each of Tables S2-S7 have the epithelial layer and cluster results separated across multiple tabs. The layer and cluster data could each be easily combined into a single tab because the “layers” column indicates already the layer or cluster. This would make these data much easier to explore for readers.

Figure 3B. As in Figure 2C, this would benefit from stratification by or annotation with the submucosa clusters in which the genes were enriched.

Line 293: This text and intro mention 3,771 DEGs, but the sum of “1,403, 1,078, and 1,041” (Line 202) is 3,522.

Figure 6A: To improve interpretation, I recommend adding the brief description of each SM to above the SM label (at an angle for space) in the correlation plots. For example, “SM1: epithelial barrier function”. Also, please italicize row labels of genus and species names. Atopobium is now Fannyhessea. It is strange that Basal, 1, 10, and 2 were specifically plotted… Seems it would be clearer to use either layers or clusters as the facets here.

SM1 (summarized as ‘epithelial barrier function’)

SM2 (‘protein production’)

SM3 genes (‘inflammation and processes occurring in the extracellular space’)

SM4 genes (‘regulation of intracellular protein transport and RNA processing’)

Line 332: The statement “The data confirm the microbiome-associated altered gene expression” implies a level of causality that is not supported by the study’s cross-sectional design. As the authors have not performed longitudinal or mechanistic experiments, the findings should be interpreted as associative rather than causal. This distinction is particularly important in microbiome research, where observational data are often used to inform clinical or public health strategies. In the context of HIV susceptibility, overstating causality may mislead readers or policymakers. I recommend revising this phrasing to reflect the correlational nature of the findings—for example, “The data reveal microbiome-associated differences in gene expression” or “The data are consistent with microbiome-associated transcriptional variation.”

Lines 341-343: Where are the data to support this? Figure 6A has plots for only Basal, 1, 10, and 2.

Lines 347-349: Where are the data to support these statements about relationships with nugent scores, age, sex work?

Throughout the discussion section, several statements imply a causal relationship between microbiome composition and host gene expression or tissue-level changes. Given the cross-sectional and observational nature of the study, the data support associations rather than causality. Phrases such as “responses,” “related to,” or “confirm” suggest mechanistic links that cannot be established without longitudinal or experimental validation. I recommend revising the discussion to consistently reflect the associative nature of the findings. This distinction is particularly important in microbiome research, where interpretations often inform clinical or public health strategies, and in the context of HIV susceptibility, where mechanistic clarity is critical. Some examples:

• “we were able to identify the spatial localization of these responses”

• “A highly diverse microbiome exhibited the largest differences in gene expression”

• “differences are related to”

Lines 361-363: The sentence “A highly diverse microbiome exhibited the largest differences in gene expression compared with microbiome groups dominated by L. crispatus, L. iners, and Gardnerella taxa” is both unspecific and potentially misleading. It is unclear whether “gene expression” refers to changes across the entire tissue, specific epithelial layers, or submucosal clusters. Given the spatial resolution of the study, I recommend specifying the anatomical or transcriptional compartments where these differences were most pronounced.

Discussion:

Lines 356–447: Throughout the discussion section, several statements imply a causal relationship between microbiome composition and host gene expression or tissue-level changes. Given the cross-sectional and observational nature of the study, the data support associations rather than causality. Phrases such as “responses,” “related to,” or “confirm” suggest mechanistic links that cannot be established without longitudinal or experimental validation. I recommend revising the discussion to consistently reflect the associative nature of the findings. This distinction is particularly important in microbiome research, where interpretations often inform clinical or public health strategies, and in the context of HIV susceptibility, where mechanistic clarity is critical. Some examples:

• “we were able to identify the spatial localization of these responses” (line 359)

• “A highly diverse microbiome exhibited the largest differences in gene expression” (line 361)

• “These differences are related to disruption of epithelial and ECM structures” (line 363)

• “The data confirm the microbiome-associated altered gene expression” (line 331)

• “ the most predominant gene expression alterations” (line 387, and “alterations” in line 389). Changes were not measures, differences were.

• “We could also confirm and now spatially localize previously presented findings, including microbiome-associated alterations of immune-related mechanisms that can facilitate HIV penetration (7,32,33) and trigger recruitment of activated CD4+ T cells to the genital mucosa, promoting HIV spread (7,34,35). (lines 440-444).

Lines 361–363: The sentence “A highly diverse microbiome exhibited the largest differences in gene expression compared with microbiome groups dominated by L. crispatus, L. iners, and Gardnerella taxa” is both unspecific and potentially misleading. It is unclear whether “gene expression” refers to changes across the entire tissue, specific epithelial layers, or submucosal clusters. Given the spatial resolution of the study, I recommend specifying the anatomical or transcriptional compartments where these differences were most pronounced.

Lines 365–367: The interpretation of “homeostatic immune activity” and “mucosal barrier integrity” in the L. crispatus group is speculative, as it is based solely on transcriptomic data. These biological states typically require protein-level or functional validation. I recommend softening the language to reflect inferred transcriptional signatures rather than confirmed immune or structural states.

Lines 379–381: The sentence “It is therefore surprising that our data indicated activated host responses throughout the entire ectocervical mucosa, which we interpreted as molecular signals driven by the cervicovaginal microbiome” overstates the novelty of the finding. Prior studies (e.g., Blaskewicz et al., 2011; Arnold et al., 2015; Anahtar et al., 2015) have shown that the apical ectocervical epithelium is permeable to macromolecules such as IgG and that microbial communities can influence immune activation beyond the luminal surface. I recommend revising this sentence to reflect that the observed transcriptional changes are consistent with known mucosal biology and to avoid implying causality.

Lines 392–423: The discussion of IgA and IgG gene expression (lines 411–423) could be better anchored in mucosal immunology literature. Torcia (2019) and Takada (2025) provide broader context for how IgA and IgG contribute to microbial clearance and immune tolerance in the cervicovaginal mucosa. Including these references would help clarify the implications of the observed transcriptional differences and better connect them to established host-microbiome interaction mechanisms.

Lines 411–423: The discussion of IgA and IgG expression and their roles in microbial modulation is informative but could be better integrated with known mucosal immunology. Consider elaborating on how these findings relate to established mechanisms of immune exclusion or tolerance in the cervicovaginal mucosa.

Lines 425-436: These appear to be written by someone else because the language regarding the associative nature of the study is appropriate. Nice!

Lines 438–439: The phrase “tissue-wide impact” should be clarified. The data show that transcriptional changes are most pronounced near the basal membrane and in specific submucosal clusters. Please specify which compartments are affected to better reflect the spatial resolution of the findings.

Reviewer #2: - It would be helpful to address limitations of the study in the discussion. For example, the inability to infer directionality of the microbial-host transcriptional associations, transcriptional findings were not further validated through other methods (i.e. protein quantification, IHC), and fungal and viral associations were not addressed, etc.

- Consider rewording the title. The study identifies transcriptional associations with vaginal bacterial composition and the directionality of these associations can not be inferred from the data – thus the microbiome may not impact host transcriptional signatures.

- Consider de-emphasizing the conclusions related to susceptibility to HIV acquisition. This was not the focus of the study, and all participants were HIV negative.

- The text mentions L. crispatus or L. jensenii in group 1 – Fig. 1A indicates L. crispatus/acidophilus. Please clarify and check for congruency with Supp. Fig. 4.

- Consider updating taxonomy where relevant. Atopobium is now referred to as Fannyhessya (PMID: 30186281)

Reviewer #3: Other concerns are relatively minor, outlined below:

Grammatical/formatting Issues:

1. Line 169: “Cluster 4 exhibited increased expression of several immunological and endothelial genes. Clusters 0 and 3 showed down regulation of genes that were more pronounced in other clusters.” Wording is vague and the second sentence is redundant with the idea of DEG. I think it can be combined with the following one and simplified.

2. Lines 180- 184- have any previous studies done transcriptomics of these tissues? GEX or spatial? Would be good to mention

3. Line 216 ref: “…including MIF and FABP5, which are linked to macrophage responses and retinoic acid metabolism”

4. Line 232 ref: “…STAT3 activator and is involved in tissue repair”

5. Line 263 ref: “ involved in cell adhesion and ECM remodeling.”

6. Lines 270-272: “ L1 was associated with upregulation of various innate immune and barrier function genes that maintain a protective and balanced immune environment”

a. It was not discussed previously in this section which genes upregulated in L1 are associated with barrier function and no reference to how they’ve been shown to maintain protective and balanced immune environment

7. No comment on how L4 was more distinct when looking at pairwise comparisons of DEG, but L3 stood out when analyzing the spatial gene modules

8. Line 349 what is a Nugent score?

9. Line 418 ref: “…facilitate bacterial clearance and immune tolerance.

10. Line 419 ref: “… IgG promotes death and clearance of bacteria through complement…”

11. Line 448-450. This is a poor final sentence to end the discussion section as it is introducing an example rather than summarizing final conclusions or future directions.

PLOS authors have the option to publish the peer review history of their article (what does this mean? ). If published, this will include your full peer review and any attached files.

**Do you want your identity to be public for this peer review?** For information about this choice, including consent withdrawal, please see our Privacy Policy .

Reviewer #1: No

Reviewer #2: No

Reviewer #3: No

**Figure resubmission:**

**Reproducibility:**



---

## [Decision Letter · Decision Letter 1]

21 Oct 2025

PPATHOGENS-D-25-01489R1

The cervicovaginal microbiome associates with spatially restricted host transcriptional signatures throughout the human ectocervical epithelium and submucosa

PLOS Pathogens

Dear Dr. Kaldhusdal,

Thank you for submitting your manuscript to PLOS Pathogens. We are pleased to inform you that in principle we will accept your manuscript; there are just a few minor corrections to be made. Please submit your revised manuscript within 30 days; when you're ready to submit your revision, log on to https://www.editorialmanager.com/ppathogens/ and select the 'Submissions Needing Revision' folder to locate your manuscript file.

We look forward to receiving your revised manuscript.

Kind regards,

Julia Oh

Academic Editor

PLOS Pathogens

Michael Otto

Section Editor

PLOS Pathogens

Sumita Bhaduri-McIntosh

Editor-in-Chief

PLOS Pathogens

orcid.org/0000-0003-2946-9497

Michael Malim

Editor-in-Chief

PLOS Pathogens

orcid.org/0000-0002-7699-2064

**Journal Requirements:**

**Reviewers' Comments:**

Reviewer's Responses to Questions

**Part I - Summary**

Reviewer #1: The authors sufficiently responded to my comments.

Reviewer #2: The authors have thoughtfully revised the manuscript to incorporate reviewer critiques. I have no further comments to address.

Reviewer #3: The authors have satisfied my concerns.

**Part II – Major Issues: Key Experiments Required for Acceptance**

Reviewer #1: None.

Reviewer #2: (No Response)

Reviewer #3: (No Response)

**Part III – Minor Issues: Editorial and Data Presentation Modifications**

Reviewer #1: • Ensure supplemental figures and figure legend numbers match.

o Fig S1: Legend is “S1 Figure. Quality control”, but provided figure labeled as “Figure S1” is “ 16S taxonomical annotation”

o For the “16S taxonomical annotation” and supplemental figure and Fig. S5: species and genus names need to be italicized. The legend should be “16S rRNA gene-based proportions of vaginal taxa” … or something to that effect.

• Figure 6: italicize species and genus labels.

Reviewer #2: (No Response)

Reviewer #3: (No Response)

PLOS authors have the option to publish the peer review history of their article (what does this mean? ). If published, this will include your full peer review and any attached files.

**Do you want your identity to be public for this peer review?** For information about this choice, including consent withdrawal, please see our Privacy Policy .

Reviewer #1: No

Reviewer #2: No

Reviewer #3: **Yes: ** Steven Bosinger

**Figure resubmission:**

**Reproducibility:**



---

## [Editor Report · Decision Letter 2]

29 Oct 2025

Dear Kaldhusdal,

We are pleased to inform you that your manuscript 'The cervicovaginal microbiome associates with spatially restricted host transcriptional signatures throughout the human ectocervical epithelium and submucosa' has been provisionally accepted for publication in PLOS Pathogens.

Best regards,

Alice Prince

Section Editor

PLOS Pathogens

Alice Prince

Section Editor

PLOS Pathogens

Sumita Bhaduri-McIntosh

Editor-in-Chief

PLOS Pathogens

orcid.org/0000-0003-2946-9497

Michael Malim

Editor-in-Chief

PLOS Pathogens

orcid.org/0000-0002-7699-2064
---

## [Editor Report · Acceptance letter]

Dear Kaldhusdal,

We are delighted to inform you that your manuscript, "The cervicovaginal microbiome associates with spatially restricted host transcriptional signatures throughout the human ectocervical epithelium and submucosa," has been formally accepted for publication in PLOS Pathogens.

Best regards,

Sumita Bhaduri-McIntosh

Editor-in-Chief

PLOS Pathogens

orcid.org/0000-0003-2946-9497

Michael Malim

Editor-in-Chief

PLOS Pathogens

orcid.org/0000-0002-7699-2064